# Cumulative advantage and citation performance of repeat authors in scholarly journals

**Kyle Siler**[1]*, **Philippe Vincent-Lamarre**[1], **Cassidy R. Sugimoto**[2], **Vincent Larivière**[1]*

**1** École de Bibliothéconomie et des Sciences de l'information, Université de Montréal, Montréal, Québec, Canada, **2** School of Public Policy, Georgia Institute of Technology, Atlanta, Georgia, United States of America

* ksiler@gmail.com (KS); vincent.lariviere@umontreal.ca (VL)

**Data Availability Statement:** The aggregated, supporting data is available on Figshare at [ https://doi.org/10.6084/m9.figshare.19358789 ]. The data are proprietary and are property of Clarivate Analytics and Leiden University. Data are available

## Abstract

Cumulative advantage–commonly known as the Matthew Effect–influences academic output and careers. Given the challenge and uncertainty of gauging the quality of academic research, gatekeepers often possess incentives to prefer the work of established academics. Such preferences breach scientific norms of universalism and can stifle innovation. This article analyzes repeat authors within academic journals as a possible exemplar of the Matthew Effect. Using publication data for 347 economics journals from 1980–2017, as well as from three major generalist science journals, we analyze how articles written by repeat authors fare vis-à-vis less-experienced authors. Results show that articles written by repeat authors steadily decline in citation impact with each additional repeat authorship. Despite these declines, repeat authors also tend to garner more citations than debut authors. These contrasting results suggest both benefits and drawbacks associated with repeat authorships. Journals appear to respond to feedback from previous publications, as more-cited authors in a journal are more likely to be selected for repeat authorships. Institutional characteristics of journals also affect the likelihood of repeat authorship, as well as citation outcomes. Repeat authorships–particularly in leading academic journals–reflect innovative incentives and professional reward structures, while also influencing the intellectual content of science.

## 1. Introduction

Cumulative advantage is a common mechanism underpinning and exacerbating social inequalities. Due to unique institutional, cultural, and personal attributes of academic professions, cumulative advantage is an especially prevalent phenomenon in science. To explain cumulative advantage in science, Merton [1] famously coined the Matthew Effect, a term denoting processes by which privileged scientists accrue further advantages and rewards solely by virtue of their status. These processes are at odds with Merton's [2] norm of universalism–the notion that scientists and their work should be judged and rewarded irrespective of their personal or social characteristics–as well as contemporary social norms regarding meritocracy

for researchers who meet the criteria for access to this dataset. To obtain the bibliometric data in the same manner as authors (i.e. by purchasing them), readers can contact Clarivate Analytics at the following URL: https://clarivate.com/products/web-of-science/contact-us.

**Funding:** Alfred P. Sloan Foundation Award Number: G-2020-12678 The funders had no role in study design, data collection and analysis, decision to publish, or preparation of the manuscript. No salary.

**Competing interests:** No competing or commercial interests.

and fairness. Academic journals are the heart of the scientific reward system, characterized by status hierarchies of publication outlets. In this system, top journals attract and develop what are believed to be the most important articles, which in turn bestow symbolic capital upon authors.

Via talent, social status and/or luck, repeat authors occupy disproportionate intellectual space and attention in top journals and academic fields. This article examines repeat authorships within academic journals–authors who publish repeatedly in the same journal–as an exemplar of the Matthew Effect in science. In particular, we analyze the prevalence of repeat authorships in various academic journals, as well the citation performance of articles written by repeat authors. Citation performance is one indicator–among others–of an article's success and usefulness in academia. Recent studies have used citation outcomes of published articles to gauge whether gatekeepers were overly permissive or harsh in evaluating certain articles [3, 4]. We apply this principle to repeat authors in academic journals. Specifically, we examine whether the citation performance of articles written by repeat authors is better or worse than contributions from debut authors. The citation performance of repeat authors can reveal evidence whether journal gatekeepers tend to be relatively harsh, permissive or neutral towards submissions from repeat authors. Using citation datasets of articles published in three leading generalist science journals, as well as 347 economics journals, we examine the citation performance of repeat authors in a variety of publishing contexts. Numerous status and professional life-course factors influence career and innovation incentives for academics, as well as signaling and gatekeeping incentives for journals. These factors will be discussed, focusing on how they might influence the prevalence and innovative impact of repeat authors in varying academic journals.

## 1.1. Article overview

First, we discuss cumulative advantage processes in science, and how they relate to repeat authors in academic journals. Then, we discuss career and life-course factors in academic careers, which exert social and intellectual influences on the work scholars produce. Repeat authors may tend to offer different innovations than debut authors, which influences their prevalence and innovative impact in academic journals. We also discuss the role journals and gatekeepers play in promoting academic ideas and careers, particularly as high-status journals exert substantial intellectual and professional influence over academic reward structures. Given the high rejection rates and competitiveness of many high-status journals, the relative prevalence of repeat authors in such journals is intellectually and professionally significant.

Using Web of Science data, we empirically identify the prevalence of repeat authorship in various types of academic journals. In particular, we focus on how journal status is related to the number of repeat authors in a journal. Then, we examine how the citation impact of published articles varies when written by repeat authors vis-à-vis debut authors. We also analyze how citation impact changes with each additional publication for the few–but significant–authors who have multiple repeat authorships in a given journal. Feedback and learning effects of successful publications are also investigated. Scholars and gatekeepers alike may be influenced by highly-cited articles with future submissions to the same journal. Thus, we examine how citation performance of an article increases the likelihood of future authorships in the same journal. We also analyze possible 'chaperone' effects [5], where previous co-authorship with high-status senior authors can bolster the careers of junior scholars. Specifically, we compare the performance of debut authors with and without previous 'chaperone' publications.

Our research provides new evidence and perspectives on the incentives, hierarchies and reward structures of modern science, as reflected through the publication system. The

prevalence and citation performance of repeat authors in academic journals reflect innovative incentives and outputs in science. This raises normative and policy issues regarding systemic costs and benefits of cumulative advantage in professional life. Cumulative advantage affects fairness and innovation in professional and creative contexts, raising normative issues regarding whether stakeholders and institutions should take actions to mitigate such processes.

## 2. Theoretical background

### 2.1. Status and cumulative advantage in science

Two main mechanisms underpin the Matthew Effect: privileged actors receive 1) more favorable evaluations and 2) increased resources [6]. Causal relationships between quality and status can interact and flow in both directions [7]. When faced with uncertainty, people often weight the social status and other ascriptive characteristics of others to inform appraisals and decisions [8]. In science, academics are more likely to invoke particularistic characteristics of authors (e.g., institutional status, gender) as decisive information under conditions of uncertainty [9–11], such as at the frontier of new scientific research [12]. Particularly in evaluative settings, academics are often influenced by the social status of authors. Numerous studies have identified that higher-status academics tend to receive more favorable evaluations [13–17].

Merton [1] posited that science was prone to generating Matthew Effects; self-fulfilling prophecies where high-status scholars accrue further rewards and cumulative advantages by virtue of their privileged status. Relatedly, intellectually conservative tendencies and incentives have also been identified in science [18–21]. Successful academics accrue power and influence, enabling leaders in scientific fields to judge academic work according to their preferred principles, in a sort of 'victor's history.' The phenomenon of preferring intellectually similar work is known as *cognitive particularism* [22]. Biases favoring cognitively proximate work or from socially close authors may have benefits. Past studies have found that evaluation quality [23] and citation impact [24–26] improve with increased social and intellectual closeness of referees. Further, academic journal editors tend to handle submissions from repeat authors more rapidly and favorably [27]. In turn, Matthew Effects in science can be partly underpinned by benign–if not rational–incentives and may sometimes generate some positive consequences for gatekeepers and broader academic fields.

Established authors may have signalling advantages with accruing citations after high-profile publications, as they have pre-existing reputations and histories to establish visibility and credibility with other scholars. When academics receive high-profile awards, their previous publications receive a boost in citations [7], which also causes intellectually proximate scholars to be crowded out of the research area [28]. Prestige-garnering publications in high-profile journals may function like similar public adornments of status on scientists. Established scholars also tend to possess professional advantages with social and intellectual networks, further helping them develop and disseminate their work. In turn, academia tends to reproduce itself in both ideas and personnel [18]. Consequently, academia usually updates orthodoxies slowly and tends to protect the status quo [29]. Paradigmatic and professional advances are often only made possible via the death or retirement of prominent scholars, opening attention and journal space for other academics, as science advances "one funeral at a time" [30]. Thus, the phenomenon of repeat authorship should be understood in part though social and intellectual advantages established scholars tend to possess.

### 2.2. Career and aging effects in science

Professional age is one factor which influences authorial strategies, goals, and cognitions in science. Cognitive skills vary–some qualities improving, others attenuating–over both

professional careers and the broader life-course [31]. In turn, people tend to reach peak career performance at different ages in different professions. Academic professions also present scientists with differing resources and incentives in their early, middle, and late careers. Accordingly, academics vary in their intellectual preferences and professional choices throughout their careers [32, 33]. Creativity and prolificness vary throughout academic careers [34, 35], as the tacit knowledge, social networks, experience, and reputations scholars develop over time all influence their published outputs.

Given advantages accrued by established academics, Merton [2] dubbed science a *gerontocracy*. A review of previous studies on scientific productivity and age found that different case studies yielded advantages for younger scholars [34], while others showed advantages for older scholars [36]. In other cases, the relationship between age and productivity is curvilinear, with advantages [37] and disadvantages [38] for mid-career scientists. Over time, academics tend to transition into different authorship roles based on seniority [39] and previous publishing success [5]. In turn, academic careers and innovation involve navigating trade-offs between *liabilities of newness* [40] vis-à-vis *liabilities of senescence* [41]. Exogenous and institutional factors influence the relationship between age and innovation in academic careers. For example, the average age of scientists making major discoveries in science is getting progressively later [42]. Academic disciplines may be changing professionally and cognitively, but increases in lab size and specialization, as well as hiring bottlenecks and declines in funding are also influencing these delays [42, 43]. Such changes in the academic opportunity structure favor older–if not also repeat–authors in academic journals.

If older or repeat authors receive more citations with later articles, this could be an indication of skill increasing over the course of the career of a scientist. Professional successes and failures influence future decision-making; effective learning from outcomes can contribute to skill improvement [44]. However, success is also conducive to increased specialization in the future, as researchers tend to exploit and expand upon successful established niches in science, as opposed to exploring new terrain [45–47]. The inverse relationship between success and exploration may influence successful scientists to be more conventional and less innovative later in their careers.

## 2.3. Skill and luck in academic careers

Through internal labor markets, as well as tenure and promotion protocol, academia winnows scientists over time. Scholars cannot accrue lengthy publication histories if not given the opportunity. In turn, longevity alone may be associated with skill in science. The infamous "publish or perish" dictum in science may privilege quantity over quality. Numerous observers have expressed concerns that some academics sacrifice quality for quantity of publications in their careers [48–50]. A recent analysis of National Academy of Sciences members found that the positive relationship between productivity and highly-cited articles can be explained solely by the fact that prolific authors produce more opportunities to have a 'hit' article [51]. Simonton's [35] Equal Odds Ratio posits that "the relationship between the number of hits and the total number of works produced in a given time period is positive, linear, stochastic, and stable."

High citation counts accrued by authors or articles often involve random, lucky, or extraneous influences. However, more productive academics are more likely to have a relatively higher proportion of highly-cited papers, suggesting that cumulative advantages play a role in the attribution of rewards [52]. Reflective of the influence of luck and serendipity on academic careers and breakthrough innovations, previous studies have found that there are random elements in academic publishing. Scientists can produce high-impact work at any juncture of

their careers [53, 54]. Peer review often involves arbitrary or random elements that develop and select published science, particularly as many highly-competitive journals have acceptance rates of less than 10% [55–57]. Similar analyses building on agent-based models corroborate the importance of luck in success in science [58, 59].

## 2.4. Case study: Academic publishing and cumulative advantage

Academic journals and their gatekeepers can both amplify and mitigate cumulative advantage in science. Our research focuses on repeat authorship within academic journals as a specific mechanism of cumulative advantage. In most social contexts, including academia, status affects evaluation. We use the context of academia to show how institutions affect cumulative advantage processes. Cumulative advantage influences the professional composition, as well as the innovative and intellectual content of science. These cumulative advantage processes underpin professional and innovative incentives for scholars and gatekeepers alike.

We use the academic discipline of economics as a case study, due to its particularly strong professional boundaries and steep intra-professional status hierarchies [60–62]. The field of economics is distinctive within the social sciences both for its heightened prestige and visibility as a discipline, as well as low levels of interaction (i.e. citations, publications, labor markets, training) with other disciplines [63–65]. These relatively strong intellectual and professional boundaries [66, 67] demarcate economics as a distinctive, autonomous academic field. In turn, the discipline and profession of economics offers a unique, competitive, hierarchical context to analyze factors that underpin cumulative advantage and innovative successes.

To complement our analysis of the discipline of economics, we also examine three prominent generalist multidisciplinary journals–*Nature*, *Science* and *Proceedings of the National Academy of Sciences* (PNAS). Using these three generalist journals as additional case studies, this enables analysis of repeat authorship in multiple contexts, including numerous different disciplinary and multidisciplinary academic fields. The economics journals provide a disciplinary context to examine repeat authors, while the three generalist journals provide an interdisciplinary context.

## 3. Methods and data

### 3.1. Data

Published articles from 347 economics journals from 1980–2017 period were retrieved from Clarivate Analytics' Web of Science, hosted at the Leiden University Centre for Science and Technology Studies (CWTS). Our data includes all journals categorized under the discipline 'Economics' in the 2017 Clarivate Journal Citation Report. Article authors were disambiguated using the method developed by Caron and van Eck [68]. The dataset includes 74,697 distinct authors based in the United States who had at least one authorship on the 154,784 identified papers, leading to a total of 244,110 author-paper combinations (or authorships). We also used a second dataset including all publications from *Nature*, *Science*, and the *Proceedings of the National Academy of Sciences* (NSP dataset) including 235,409 unique authors having contributed 134,030 articles, for a total of 549,175 authorships. We conservatively limited the analysis to United States authors in order to restrict potential influences caused by international differences. Academic publishing cultures, incentives and dynamics vary by country, so limiting analysis to United States-based authors ensures a relatively homogenous collection of scholars to analyze. Moreover, United States-based authors account for the majority of economics articles, as well as of articles in *Nature*, *PNAS* and *Science* in our dataset.

### 3.2. Dependent variable: Citations received

Since we are analyzing factors conducive to article visibility and the diffusion of ideas in science, we use citations as an indicator of academic influence and attention. While citations are not necessarily a signal of inherent academic quality, they are signaling which articles receive attention, prominence, and usage in academic fields [69]. Since citations tend to be exponentially distributed, with a few articles possessing extremely large values on the right tail of the distribution [70, 71], the logarithm of citations was taken following this equation:

$$c_i = \log(a_i + 1) + 1.$$

Where $a$ is the number of citations received by each article. Then, we standardized the citations received by journal per year, following:

$$z_i = \begin{cases} \dfrac{c_i - \mu_{JY}}{\sigma_{JY}} & \text{if } \sigma_{JY} > 0 \\ 0 & Otherwise \end{cases}.$$

Where $\mu_{JY}$ and $\sigma_{JY}$ are the mean and standard deviation log citations $c$ of all articles published in each year and journal.

In turn, the dependent variable in this study is the z-score of the logarithm of citations received per year and journal for each published article. This transformation made it so that publications are compared to others published in the same year and journal.

### 3.3. Independent variables

In order to measure repeat authorship, we compiled for each author on the byline of each article, whether that article represented the $1^{st}$, $2^{nd}$, $3^{rd}$,... $N^{th}$ article published in that journal by that author as the senior contributor. For each article, seniority is attributed to the author who published the most papers in the same journal prior to the publication of the manuscript. Many articles in our dataset have multiple authors. Authors with numerous different social and demographic characteristics can co-exist on the byline of the same article. For the purposes of our research, we assume that credit and attention will tend to focus on the most 'distinguished' author on each co-authored paper.

Like in most academic disciplines, there is a hierarchy of journals in economics. In in economics, this hierarchy is especially pronounced. Publishing in "Top Five" journals (*American Economic Review*, *Econometrica*, *Journal of Political Economy*, *Quarterly Journal of Economics*, *Review of Economic Studies*) carries enormous intellectual and professional influence in economics [72]. We use percentile ranks (0–50, 50–75, 75–90, 90–99) by Journal Impact Factors (JIFs) and elite status ("Top 5" journals) as empirical measures of journal prestige. Notably, due to their special status in the economics profession, "Top 5" journals were analyzed separately from the top JIF decile. The rank of journals was obtained by ranking all journals for each year where they were each active between 1980 and 2017. We then computed an average rank for each journal, and the distribution of average ranks was split into percentile rank categories and "Top 5" journals. It is expected that journals with higher impact factors will inherently generate more citations for published articles. In turn, it is necessary to include JIFs as a control variable while using total citations as a measure of scientific influence.

### 3.4. Mixed-effect models

Linear mixed-effect models were used to account for repeated measures between journals and authors. We do not report *p*-values focusing instead on the coefficients of the models. We used

the lme4 package [73] in R to fit the mixed-effect models, and the arm package [74] to extract the standard error of the model coefficients.

### 3.5. Author-level repeat authorship

We modelled the citation score (the natural logarithm of citations received, normalized by year and journal) as the dependent variable. For the economics dataset, we modelled both the author and the journals as random effects and obtained random coefficients for the JIF ranks. The publication order was considered as a nominal variable because of the non-linear nature of the relationship. In other words, a separate coefficient is estimated for each JIF rank/order level. We did not include an intercept in the model in order for the polarity of the coefficients to be interpretable (with an average citation score of 0). In contrast to the economics dataset, for the NSP data the journals were not included as random effects, but as random coefficients instead of the JIF rank.

### 3.6. Probability of repeat authorship

We used a logistic mixed-effect model with a binary dependent variable indicating if an author published in the senior position again in the same journal in the future. Like the previous model, the economics dataset had the journal as a random effect and the NSP dataset had the journals as random coefficient instead of the JIF percentiles. This model included the authors as a random effect. However, this model predicted the future publication based on the citation score of the previous publication of the author as a senior author. We discretized the citation score in four quartiles, which were used as random coefficients with the JIF rank (economics) or journal (NSP).

### 3.7. Influence of chaperones

Sekara et al. [5] identified the "chaperone effect" in academic publishing, where co-authoring with prominent senior authors is conducive to transitioning to senior authorship positions in the future. In order to establish the impact of publishing with a more senior author prior to publishing their first senior author publication, we used a mixed-effect model with the citation score of the first senior author publication as a dependent variable. The model uses the journal and authors as random factors for the economics dataset, and the authors for the NSP dataset. It then used the JIF rank (or journal for NSP) and whether or not the author has published with a more senior author before (i.e. previous co-authorship with a chaperone in a focal journal) as random coefficients.

## 4. Results

Fig 1 shows the cumulative distribution functions of repeat authorships by JIF percentage and journal (S1 and S2 Tables). Institutional characteristics appear to influence the prevalence of repeat authors in journals. Higher-status journals in economics tend to have more repeat authors. This is a notable finding given the intense competitiveness and selectivity of elite economics journals (see [3, 72]). *PNAS* has more repeat authors than *Nature* or *Science*, perhaps reflecting the influence of institutional membership with the National Academy of Sciences and concomitant publishing opportunities, especially with articles contributed to the journal by members of the National Academy of Sciences (see [75]).

Fig 2 presents the crossed linear mixed effect model without an intercept, showing expected citation differences depending on the repeat author status of senior authors of published articles. Author-level analysis suggests diminishing returns to repeat authorships. Put differently,

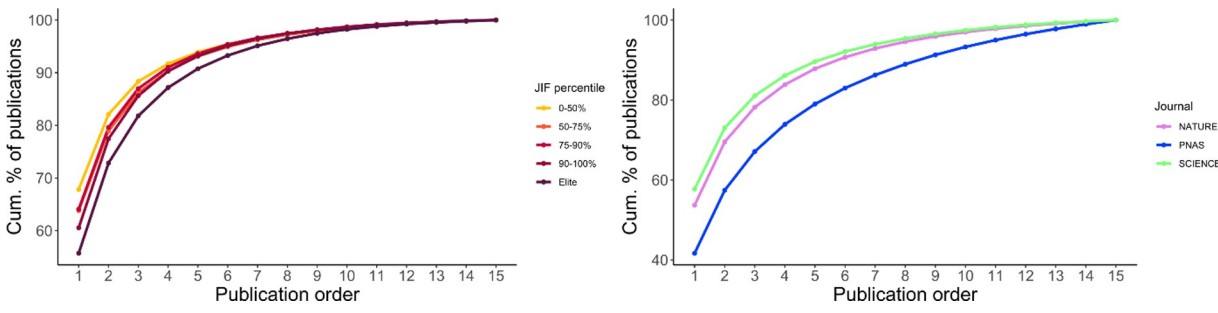

**Fig 1.**

senior repeat authors tend to produce their most highly-cited work with their debut contribution, then citations gradually decline with each subsequent publication in the same journal. There were no major appreciable differences in tendencies between the three generalist journals in the NSP dataset. However, the impact decline of repeat authors with each subsequent publication is much stronger in elite economics journals vis-à-vis all other economics journals.

Fig 3 illustrates results from the article-level of analysis (S3 and S4 Tables), which contrasts with the author-level analyses illustrated in Fig 2. Each data point refers to the average citation and standard error for the JIF (economics) or journal (NSP). On the whole, articles receive more citations with increases in repeat authorship. This supports the hypothesis that journals tend to benefit from publishing repeat authors, conditional on previous citation performance. Even if there are declines in citations within the careers of publishing authors, repeat authors can still be advantageous for journals because there is a positive correlation between consecutive publications (S1 Fig), assuming that journals select repeat authors are influenced by previous performance. In economics, there do not appear to be major status or institutional differences between journals with this general trend. *PNAS* exhibits a relatively weaker citation advantage for repeat authors than *Nature* and *Science*. This could be related to the finding that repeat authorships are less common in *Nature* and *Science* than in *PNAS*.

Fig 4 illustrates a possible mechanism underpinning differential citation performance of repeat authors in different journals (S7 and S8 Tables). Journals vary in the degree to which the previous citations accrued by an author affects the likelihood of future (repeat) authorship. The model in Fig 4 uses the citation score of the previous publication of an author as a predictor of whether or not they would publish a subsequent article in the same journal. Once again, we used a crossed logistic mixed-effect model where authors and journals have random intercepts. We binned the citation score in quartiles, with the lowest quartile (4th) as a reference category. The positive slope observed in the log-odds of repeat publication shows that the higher

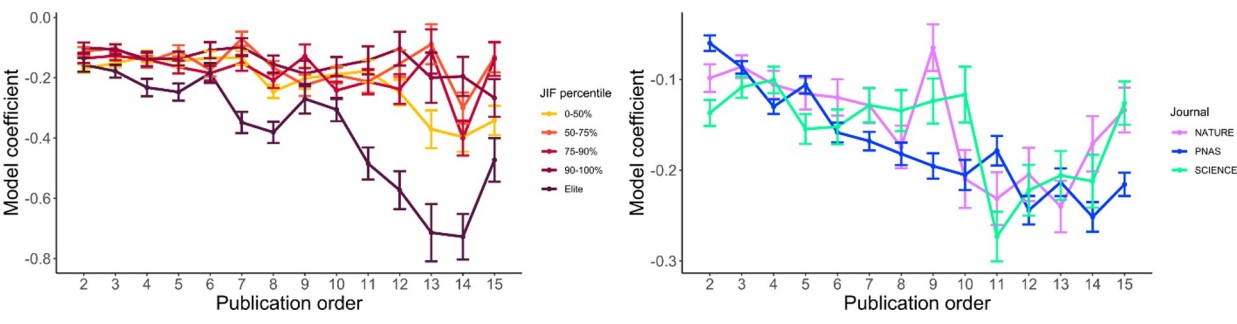

**Fig 2.**

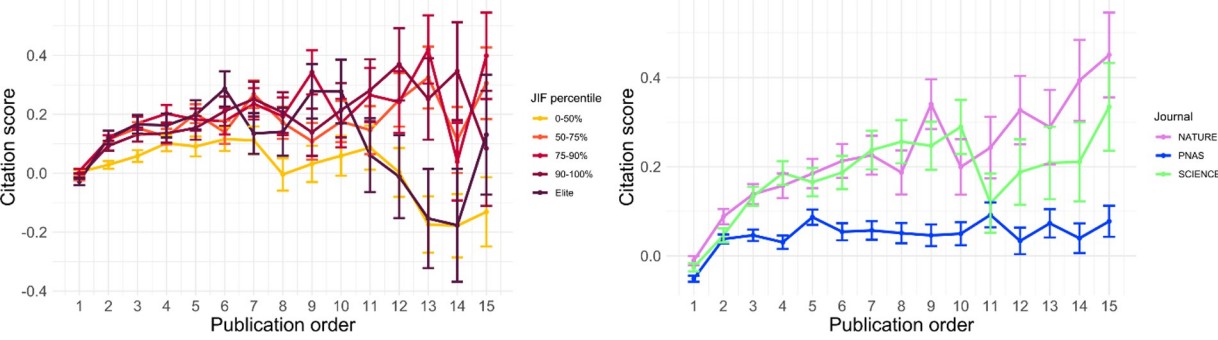

**Fig 3.**

quantile of the citation score of the previous publication, the higher the odds than an author will publish in the same journal again. The slopes are similar for journals of varying status levels, with the exception of lower-status (bottom 50%) economics journals, which are slightly less sensitive to the previous citation score overall. Analogously, *PNAS* appears less sensitive to the previous citation scores of repeat authors than *Nature* or *Science*.

Fig 5 illustrates effects of 'chaperones' on citation performance as another facet of repeat authorship. We took the first publication of every author in a given journal as a senior author. In co-authorship cases of authors with identical past experience, we attributed the senior position randomly. We then assigned the authors in two groups depending on whether they published with a more senior author in the journal. Across all of the journals in our dataset, authors without chaperones tended to receive fewer citations overall. In economics, the citation penalty of lacking a chaperone was strongest in higher-status journals. Analogously, *PNAS* exhibited slightly greater citation underperformance for articles without chaperones than *Nature* and *Science*.

## 5. Discussion

Our findings suggest mixed incentives associated with repeat authors. Although the citation impact of articles from repeat authors steadily declined with each additional published article in the same journal, there are still incentives for journals to publish repeat authors. Even if individual repeat authors experience citation declines with each additional publication, they still tend to garner above-average citation counts within that particular journal. Thus, there appear to be incentives for journals and gatekeepers to publish repeat authors, especially when

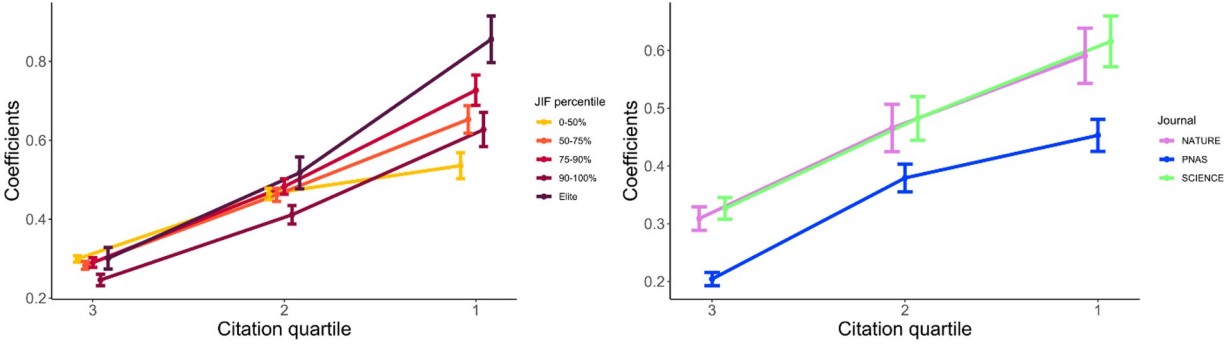

**Fig 4.**

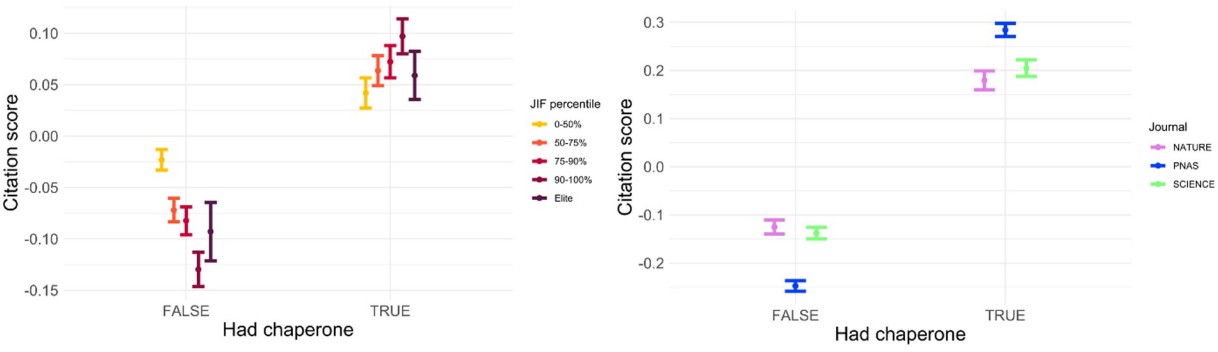

**Fig 5.**

those authors garnered high citation counts with previous articles. These creative incentives exist in contexts beyond academia. Analogously, the film industry is prone to preferring sequels over new franchises. Risk-averse studios prefer the security of leveraging the success of a proven 'parent' brand over trying new innovations. Much like the repeat academic authors in our research, although movie sequels are usually less profitable than predecessor films, they still tend to financially outperform most new contributions [76].

Institutional characteristics of journals influence the prevalence of repeat authors, as well as the citation outcomes of those repeat authors. The hierarchical academic field of economics exhibited varying trends and outcomes regarding repeat authorship, depending on the status of the journal. In particular, the elite "Top Five" economics journals–which hold substantial professional and intellectual influence–exhibited contrasting results with other economics journals. Even though publishing in those five economics journals is extremely competitive, they published more repeat authors relative to lower-status journals. Whether this is due to skill, luck and/or social connections of those repeat authors is an open question. Repeat authors in elite economics journals exhibited larger citation declines with repeat authorships than in other journals. This suggests that in elite economics journals, repeat authors make their largest impacts with their debut article. However, despite this apparent benefit of new contributors, debut authors without co-authoring 'chaperones' who have previously published in the journal were relatively less-cited in higher-status economics journals. With the general-ist journals in our study, *PNAS* exhibited some different trends vis-à-vis *Nature* and *Science*. *PNAS* had relatively more repeat authors than *Nature* and *Science*, repeat authors had a smaller citation advantage, and appeared to be less-sensitive to the previous performance of repeat authors in the journal. These differences are likely at least in part due to *PNAS*'s institutional links to the National Academy of Sciences and unique peer review structure (see [75]). In sum, strategic and cultural characteristics of academic publishing institutions affect representation and innovation.

## 5.1. Risk, reward and editorial decision-making

Editors and journals may rationally prefer articles written by repeat authors out of risk-aversion and/or a reasonable belief that repeat authors tend to achieve relatively better innovation and citation outcomes. Given our results–which showed that repeat authors generally receive more citations than debut authors–this is likely an additional incentive compelling editors and gatekeepers to harbor preferences for publishing repeat authors. Leaders tend to be more cognizant of downside risk than upside risk [77]. The uncertainty of the scientific research frontier [12] is also conducive to decision-making challenges. People tend to rely on heuristics–simple

rules and schemas–to inform decisions when faced with uncertainty [78]. These decisions and heuristics can be informed by otherwise irrelevant or arbitrary social and personal characteristics [8]. In turn, uncertainty can breed risk-aversion and preferences for the intellectual and professional status quo, particularly when more certain options are present [79]. Confirmation bias has been documented in science, where evaluators prefer work that reflects the status quo [80]. In short, there are numerous social-psychological reasons why editors and journals may prefer repeat authors.

Our results also raise normative and empirical issues regarding whether academic journals should prefer repeat authors, from both fairness and innovation perspectives. Alternatively, should journal gatekeepers take action to include more debut and less-experienced authors? Since academics tend to prefer to cite high-status authors and studies [7, 11], are redistributive policies and actions warranted to counteract such biases? In some organizational contexts, rewards are redistributed to less-privileged actors in efforts to offset cumulative advantage processes [81]. For example, in a study of four high-status economics journals, Card and Della-Vigna [3] found that more prolific authors tended to be more highly-cited, leading them to conclude that editors at such journals judge submissions from high-status authors relatively stringently. Journal peer review can potentially amplify or mitigate cumulative advantage processes and hierarchies in science.

## 5.2. Learning, feedback and editorial preferences

Learning is another factor that influences the success of repeat authors in peer review, as well as the scientific output of those authors. The impact of citation feedback on institutional learning is especially important given our findings that journals appear to select repeat authors in part on previous citation performance. Journal editors learn from their experiences interacting with authors in the peer review system [57]. For authors, experience with the peer review system in a given journal–whether as an author or peer reviewer–helps develop tacit knowledge to successfully navigate that system in the future. Since innovators tend to repeat or emulate successful outcomes, this underpins incentives to focus on exploiting successful niches, instead of exploring new terrain [45, 47]. Exploitation of normal science might be a safer choice for authors and gatekeepers alike but tends not to generate breakthrough innovations and paradigm shifts [29, 82]. Experience and positive feedback might be valuable for academics, by improving their propensity to successfully navigate peer review and publish their work in preferred outlets. Paradoxically, these learning processes and incentives might also undermine strategies and preferences for generating high-impact work. Repeat authorship increases the likelihood of redundancy in both authors and academic output. If scientific innovation is a matter of randomness or volume of ideas produced [35, 51, 54], then producing similar ideas will reduce the odds of a breakthrough innovation.

## 5.3. Status and incentives in academic publishing

Repeat authorship reflects innovative incentives within scientific careers, which has broader consequences for field-level innovation. While learning theory posits that success results in a narrowing of subsequent work [45, 47], accrued academic capital may be mutable and deployed in numerous ways. For example, after receiving the Fields Medal–the most prestigious prize in the field of mathematics–many mathematicians began to "play the field" and engage with numerous new research areas, at the expense of short-term productivity [83]. Legitimacy and scientific status can be transferable within and between subfields. Depending on the context, processes of cumulative advantage–or Matthew Effects–can give repeat authors

latitude to publish similar work. On the other hand, Matthew Effects can also grant high-status academics latitude to publish work on new topics with a modicum of legitimacy.

In recent years, high-status journals in economics have consolidated increasing influence over article citation outcomes. Regardless of whether article quality and/or journal status are influencing these changes, this further underpins competitive incentives to publish in high-JIF journals. Publishing in "Top Five" economics journals is extremely competitive. As of 2017, acceptance rates in elite economics journals had declined to between 2.5% (*Quarterly Journal of Economics*) and 5% (*American Economic Review*) [3]. Whether it can be explained by skill, luck and/or social connections, the persistence of the phenomenon of repeat authorship in these intensely competitive journals is notable. More broadly, scientific incentives for scholars and innovation trajectories–particularly in regards to where to attempt to publish research– are influenced by disciplinary trends and cultures. This trend of increasing concentration of influence in leading journals runs counter to most other fields in contemporary science, which are instead exhibiting trends of decreased concentration of citations in top journals [84].

## 6. Conclusion

Repeat authors are especially influential and important in science. Particularly in high-status journals, repeat authors exert disproportionate influences on disciplinary agendas. Despite the crowding and competitiveness associated with publishing in high-status journals, such journals were relatively more prone to publishing repeat authors. In our case study of economics, higher-status journals were relatively more conducive to repeat authorship. Further investigating the relationship between institutional status and cumulative advantage is a matter for future research, both inside and outside of academic contexts.

Our research also suggests that repeat authorship offers different incentives to journal gatekeepers and academics. In terms of citation impact, journal gatekeepers benefit from publishing repeat authors, especially when gatekeepers select repeat authors based on previous citation performance. In contrast to the apparent benefits of publishing repeat authors in general, within the individual careers of scientists, citation impact steadily declines with each repeat authorship. Declining citation impact with repeat authorships also suggests costs of trading exploratory for exploitative innovation strategies. Our results also suggest a potential downside of the Matthew Effect in academic publishing. Preferring repeat authors may be a risk-averse decision-making strategy for journal gatekeepers dealing with the uncertainty of appraising and choosing the most meritorious science to publish. However, these cumulative advantage incentives and processes may also present risks of undermining innovation and diversity in science, if not also professional norms of meritocracy.

## Supporting information

**S1 Table. Cumulative distributions of repeat authors for economics journals.**
(DOCX)

**S2 Table. Cumulative distributions of repeat authors for *Nature*/*Science*/*PNAS*.**
(DOCX)

**S3 Table. Author-level coefficients and standard error of citation impact by repeat authorship for economics journals.**
(DOCX)

**S4 Table. Author-level coefficients and standard error of citation Impact by repeat authorship for *Nature*/*Science*/*PNAS*.**
(DOCX)

**S5 Table. Average citation impact by publication order for economics journals.**
(DOCX)

**S6 Table. Average citation impact by publication order for *Nature/Science/PNAS*.**
(DOCX)

**S7 Table. Effects of previous citation performance on likelihood of future repeat authorship for economics journals.**
(DOCX)

**S8 Table. Effects of previous citation performance on likelihood of future repeat authorship for *Nature/Science/PNAS*.**
(DOCX)

**S9 Table. Effect of 'Chaperone' status on citation performance for economics journals.**
(DOCX)

**S10 Table. Effect of 'Chaperone' status on citation performance for *Nature/Science/PNAS*.**
(DOCX)

**S1 Fig.** Correlation between citation score of consecutive publications for economics journals (left) and *Nature/Science/PNAS* (right). Each plot includes every pair of consecutive publications in the same journal for the same senior author.
(DOCX)

## Acknowledgments

The authors thank seminar participants at the University of Arizona Department of Sociology, the University of Strasbourg Department of Economics and the Utrecht University Innovation Studies Group for helpful feedback on previous versions of this article.

## Author Contributions

**Conceptualization:** Kyle Siler, Philippe Vincent-Lamarre, Cassidy R. Sugimoto, Vincent Larivière.

**Data curation:** Kyle Siler, Philippe Vincent-Lamarre, Cassidy R. Sugimoto, Vincent Larivière.

**Formal analysis:** Kyle Siler, Philippe Vincent-Lamarre, Cassidy R. Sugimoto, Vincent Larivière.

**Investigation:** Kyle Siler, Philippe Vincent-Lamarre, Cassidy R. Sugimoto.

**Methodology:** Kyle Siler, Philippe Vincent-Lamarre, Cassidy R. Sugimoto, Vincent Larivière.

**Project administration:** Kyle Siler, Cassidy R. Sugimoto, Vincent Larivière.

**Validation:** Vincent Larivière.

**Visualization:** Philippe Vincent-Lamarre, Vincent Larivière.

**Writing – original draft:** Kyle Siler, Philippe Vincent-Lamarre, Cassidy R. Sugimoto, Vincent Larivière.

**Writing – review & editing:** Kyle Siler, Philippe Vincent-Lamarre, Cassidy R. Sugimoto, Vincent Larivière.

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
