## [Decision Letter · Decision Letter 0]

16 Aug 2021

PONE-D-21-19824

Cumulative Advantage and Citation Performance of Repeat Authors in Scholarly Journals

PLOS ONE

Dear Dr. Siler,

Thank you for submitting your manuscript to PLOS ONE. After careful consideration, we feel that it has merit but does not fully meet PLOS ONE’s publication criteria as it currently stands. Therefore, we invite you to submit a revised version of the manuscript that addresses the points raised during the review process.

Please consider carefullly all the claims suggested by the reviewers. I would like to highlight that your paper seems not to respect the data availability policy of PLOS ONE. The revised version should be compliant with PLOS data policy as stated here: https://journals.plos.org/plosone/s/data-availability. 

We look forward to receiving your revised manuscript.

Kind regards,

Alberto Baccini, Ph.D.

Academic Editor

PLOS ONE

3.  Thank you for stating the following in the Competing Interests/Financial Disclosure * (delete as necessary) section:

“Sloan Foundation Grant G-2020-12678”

We note that you received funding from a commercial source: Sloan Foundation Grant

4. Please include captions for your Supporting Information files table at the end of your manuscript, and update any in-text citations to match accordingly. Please see our Supporting Information guidelines for more information: http://journals.plos.org/plosone/s/supporting-information.

5. Please note that supplementary tables (should remain/ be uploaded) as separate "supporting information" files"

Reviewers' comments:

Reviewer's Responses to Questions

**Comments to the Author**

1. Is the manuscript technically sound, and do the data support the conclusions?

Reviewer #1: Partly

Reviewer #2: Partly

2. Has the statistical analysis been performed appropriately and rigorously? 

Reviewer #1: Yes

Reviewer #2: Yes

3. Have the authors made all data underlying the findings in their manuscript fully available?

Reviewer #1: No

Reviewer #2: Yes

4. Is the manuscript presented in an intelligible fashion and written in standard English?

Reviewer #1: Yes

Reviewer #2: Yes

5. Review Comments to the Author

Reviewer #1: This manuscript presents a bibliometric study on cumulative advantages in scholarly journals, which compared top journals in economics with some of the most prestigious ‘general’ journals (PNAS, Nature and Science). The manuscript is well written with a well-elaborated theoretical background section that reconstructs the debate on status and career in science.

While in general I like the paper, there are weaknesses in this study on which authors could improve: 1) a not-so well defended choice of the field selection in their research design; and (2) the excessive generalisation of their findings.

As regards the first point, stating that the field of economics is autonomous and has peculiarities is not sufficient to justify a comparison with PNAS, Science and Nature. In many fields, there’s a concentration of attention towards a restricted number of prestigious top journals (e.g., sociology, management etc.). In my opinion, the more interesting difference between economics journals and the three generalists (and other similar fields for top concentration of journals) is that economics is an incremental, path-dependent discipline with a strong theoretical mainstream (neo-classical economics, DSGE models), which the top journals are instrumental to defend and isolate (à la Lakatos). Economics is not about revolutionary discoveries, novelties, competition for the frontier of truth like Science and Nature. In economics, there is also still the cult of solo-authored manuscripts in top journals, which – especially at the beginning of the career – are used to determine the tenure track (similarly in sociology and management). I would discuss these differences to defend the design better. BTW, this seems to me also very much helping authors to discuss their results.

As regards the design, it’s unclear to me if authors in their data collection used the same time window in the economics and non-economics sample (1980-2017).

Why did they restrict their attention to US authors?

It’s also not crystal clear if they have used the top journals in 2017 and went back to re-calculate their IF for each year back to 1980. I assume that the top journals are relatively stable over years and so this is reasonable. But, still, not clear in the text if they controlled for possible top journals out and in in such a long temporal sequence.

Another point is seniority. Obviously it is not easy to estimate a scientist’s seniority with these sample numbers, data and time window. However, seniority could have been estimated by considering the first year of publication in Scopus rather than from the author position. Perhaps, some discussion on how to improve these measurements could be added in the closing section, which now is too concise and does not include a proper study limitation section.

As regards to conclusions/findings, the text often conveys the message that journals and editors can strategically decide on manuscripts depending on author prestige. What about peer reviewers? I have the impression that authors under-evaluate the complexity (and the distributed nature) of decision-making in academic journals. This should be discussed.

The sentence on page 19: “Cumulative advantage processes are linked to institutional properties and policies” is obscure. It seems to me a general statement that evokes certain links between findings and previous research or theories about science as an institutional system without specifying them clearly.

This also holds for some part of the discussion in 5.3 about the fact that repeat authors could have advantages of “latitude” to publish work in difrerent journals, perhaps across sub-specialities. This is – again – a claim that is not supported by evidence and data here, as authors have concentrated their attention to repeat authors in the same disciplinary journals, not across areas of research.

I personally found the part on cognitive bias in decision-making relatively disconnected from the rest and the explanation. At the end of the day, authors did not have data on editorial decisions and could not estimate whether cumulative advantages from seniors in economics journals are due to decision-making bias (i.e., 5.1 Sect).

Furthermore, when they link these outcomes to the fact that this can compromise innovation, again, it’s a strong claim that is not supported by their evidence. I would suggest to tone-down the final part as in my opinion many claims on these findings cannot be supported by this research design.

As regards to conclusions, the sentence in which authors suggest that journals would benefit from publishing repeat authors seems to be not supported by their evidence. Paradoxically, this would mean that top journals in economics and PNAS should have higher IF than Science and Nature, which seem less prone to publish repeat authors. This could be supported if authors would compare journals that are more or less prone to publish repeat authors in the same field and show that the former have high IF than the latter.

Furthermore, the statement in the last sentence of the closing section, i.e., that repeat authors compromise diversity and innovation, should be toned down. Do repeat authors publish preferably sloppy science or innovative research? Are authors suggesting that the marginal negative returns of repeat publications in terms of citations mean that these papers did not constitute innovative research? Plese, try to link these conclusions to your research more clearly.

As regards to the background section, is there an example of cultural environment in all social evolution in which individual learning does not require the capacity of identifying role models and relying on their signal/example? It could appear a purely provocative/rhetorical question. However, how could we even imagine a competitive institutional system based on continuous learning and complex incremental paths (scientists compete for extending the frontier of knowledge via priority rewards, to cite Merton who is central in this paper) without cumulative advantages? The authors make a good point in discussing learning and feedback. However, it seems to me that cumulative advantages due to bias and cumulative advantages due to learning and specialisation provide two very much alternative explanations, whereas it seems to me that authors implicitly but straightforwardly take the first line when discussing their findings.

Figures 1 and 2: the meaning of the distribution in the axis of the publication order is unclear. Please, add a legend.

Figure S1: the scale of axes must be kept similar or the comparison between the two plots is non-intuitive

Tables S1 and S3. I would add an explanation of the range of journals in the legend.

Reviewer #2: This paper provides some very interesting new descriptive statistics on the prevalence and correlates of repeat publishing in economic and general-science journals. My only concern with the paper as currently written is related to the discussion and conclusion sections of the paper, which often make suppositions that read slightly more dramatic than I believe is warranted given the authors results.

I think the authors put it best themselves when they say on pg. 18: "Whether this is due to skill, luck and/or social connections of those repeat authors is an open question."

However, the discussion and, in particular, the conclusion section is written with a rather negative tone highlighting many potential negative consequences of the patterns in the data. As the authors note, while there is evidence of decreasing citations for repeat publishers, those same papers often receive more citations than the other papers in the same journal (by non-repeat publishers). So if editors are maximizing citations, this strategy seems rational. And of course, as the authors note as well, citations may not fully capture important dimensions of paper quality (e.g., introduction of new, diverse viewpoints). But this also implies that the decline in citations amongst repeat-publishers could reflect the repeat-publishers branching out into new domains or research questions, in which case this pattern could still be a sign of "good" strategies by authors and editors. I don't mean to say that the authors' conclusions are wrong, they are absolutely reasonable hypotheses that could rationalize the observed data. My point is to say that there are many other hypotheses (that don't require negligence or biases) that can also rationalize these patterns. Thus, I would appreciate a more even handed approach to the writing in these final sections.

These remarks are why I have listed this paper as "partly" supporting the authors conclusions. I think a more balanced discussion of what could be generating these patterns in the data would be worthwhile and convince me that the data more fully supports the hypotheses put forward in the discussion and conclusion sections.

6. PLOS authors have the option to publish the peer review history of their article (what does this mean?). If published, this will include your full peer review and any attached files.

Reviewer #1: No

Reviewer #2: No

---

## [Author Response · Author response to Decision Letter 0]

18 Oct 2021

Dear Dr. Baccini,

Please find below our replies to the referee’s comments and suggestions. We appreciate their careful attention, which helped us improve the manuscript. For easier readability via the PLOS ONE editorial site, I will place two stars before each of our responses.

Thank you,

Kyle Siler

Reviewer #1: This manuscript presents a bibliometric study on cumulative advantages in scholarly journals, which compared top journals in economics with some of the most prestigious ‘general’ journals (PNAS, Nature and Science). The manuscript is well written with a well-elaborated theoretical background section that reconstructs the debate on status and career in science.

While in general I like the paper, there are weaknesses in this study on which authors could improve: 1) a not-so well defended choice of the field selection in their research design; and (2) the excessive generalisation of their findings.

As regards the first point, stating that the field of economics is autonomous and has peculiarities is not sufficient to justify a comparison with PNAS, Science and Nature. In many fields, there’s a concentration of attention towards a restricted number of prestigious top journals (e.g., sociology, management etc.). 

** Absolutely. That hierarchy is especially pointed in economics (as per the Heckman, Fourcade article we cited).

In my opinion the more interesting difference between economics journals and the three generalists (and other similar fields for top concentration of journals) is that economics is an incremental, path-dependent discipline with a strong theoretical mainstream (neo-classical economics, DSGE models), which the top journals are instrumental to defend and isolate (à la Lakatos). Economics is not about revolutionary discoveries, novelties, competition for the frontier of truth like Science and Nature. 

In economics, there is also still the cult of solo-authored manuscripts in top journals, which – especially at the beginning of the career – are used to determine the tenure track (similarly in sociology and management). I would discuss these differences to defend the design better. BTW, this seems to me also very much helping authors to discuss their results.

** Our data shows that most manuscripts in Econ are now multi-authored. We agree, however, that authorship in Econ is quite peculiar. Heather Sarsons’ recent paper (published in the Journal of Political Economy) on gender norms in economics co-authorship shows how important co-authorship is in contemporary economics.

As regards the design, it’s unclear to me if authors in their data collection used the same time window in the economics and non-economics sample (1980-2017).

** Yes, the 1980-2017 window is identical. We mention this in Section 3.3.

Why did they restrict their attention to US authors?

** We limited the analysis to US authors in order to control for country. Publishing dynamics vary by country, and we wanted to have a set of authors that is relatively homogeneous. Moreover, US authors account for the majority of econ papers as well as of papers in Science / Nature and PNAS throughout the period. 

It’s also not crystal clear if they have used the top journals in 2017 and went back to re-calculate their IF for each year back to 1980. I assume that the top journals are relatively stable over years and so this is reasonable. But, still, not clear in the text if they controlled for possible top journals out and in in such a long temporal sequence.

** The top journals (American Economic Review, Econometrica, Journal of Political Economy, Quarterly Journal of Economics, Review of Economic Studies) and deciles of journals are the same throughout the period. We used 2017 JIFs. While status hierarchies in economics are relatively stable historically, journals on the border between quartiles (and thus, continually oscillating) would make analysis difficult without using a constant time period for JIFs.

Another point is seniority. Obviously it is not easy to estimate a scientist’s seniority with these sample numbers, data and time window. However, seniority could have been estimated by considering the first year of publication in Scopus rather than from the author position. Perhaps, some discussion on how to improve these measurements could be added in the closing section, which now is too concise and does not include a proper study limitation section.

** Seniority is not based on author position but, rather, by the number of publications previously written in the journal. In other words, we defined the senior author as being the author who has published the highest number of papers in that journal.

As regards to conclusions/findings, the text often conveys the message that journals and editors can strategically decide on manuscripts depending on author prestige. What about peer reviewers? I have the impression that authors under-evaluate the complexity (and the distributed nature) of decision-making in academic journals. This should be discussed.

** Peer reviewers tend to be double-blind. So, they are much less likely to be aware of the identities of the authors. However, they still may harbor preferences for more conservative work. We added some text referring to the complexity of academic evaluation.

The sentence on page 19: “Cumulative advantage processes are linked to institutional properties and policies” is obscure. It seems to me a general statement that evokes certain links between findings and previous research or theories about science as an institutional system without specifying them clearly.

** Agreed. This sentence has been deleted.

This also holds for some part of the discussion in 5.3 about the fact that repeat authors could have advantages of “latitude” to publish work in difrerent journals, perhaps across sub-specialities. This is – again – a claim that is not supported by evidence and data here, as authors have concentrated their attention to repeat authors in the same disciplinary journals, not across areas of research.

** This is not what we suggested. We cited other relevant research showing that high-status academics are able to use their status in one domain to expand into new fields and areas of expertise. This does not necessarily imply that they are moving to different disciplines or journals. Moreover, economics has much less interaction with other disciplines in the social sciences. We cited Truc et al.’s (2021) recent work that shows just how insular economics is in its citation practices, especially in the “Top 5” journals. Even if other disciplines cite economists, economics is fairly insular in its citation distribution. Therefore, for most of the community, there is no “outside”; stopping to publish in Econ journals means stopping publishing, period. 

I personally found the part on cognitive bias in decision-making relatively disconnected from the rest and the explanation. At the end of the day, authors did not have data on editorial decisions and could not estimate whether cumulative advantages from seniors in economics journals are due to decision-making bias (i.e., 5.1 Sect).

** Indeed, we do not have data on editorial decisions. This discussion aims at speculating about the potential mechanisms; we added some references that support those explorations. We cite the recent book Secrets of Economics Editors (Szenberg and Ramrattan, 2014) that reveals that issues of cognitive bias are common among editors, although it’s rarely openly discussed.

Furthermore, when they link these outcomes to the fact that this can compromise innovation, again, it’s a strong claim that is not supported by their evidence. I would suggest to tone-down the final part as in my opinion many claims on these findings cannot be supported by this research design.

** We toned down this part. We tried not assert “fact” but, rather, the possibility that preferring established academics results in intellectually conservative outcomes, especially given prevailing theories in organization science and the sociology of science about innovation and learning.

As regards to conclusions, the sentence in which authors suggest that journals would benefit from publishing repeat authors seems to be not supported by their evidence. Paradoxically, this would mean that top journals in economics and PNAS should have higher IF than Science and Nature, which seem less prone to publish repeat authors. This could be supported if authors would compare journals that are more or less prone to publish repeat authors in the same field and show that the former have high IF than the latter.

** Our data actually supports this claim: when analyzing within-journal performance repeat authors have higher citation rates than non repeat authors. However, the more “repeats”, the lower this effect; we observe a decline in the citation performance of repeat authors within subsequent publications. We tweaked the sentence to make this clearer.

Furthermore, the statement in the last sentence of the closing section, i.e., that repeat authors compromise diversity and innovation, should be toned down. Do repeat authors publish preferably sloppy science or innovative research? Are authors suggesting that the marginal negative returns of repeat publications in terms of citations mean that these papers did not constitute innovative research? Plese, try to link these conclusions to your research more clearly.

** We toned down the language in this final sentence in order to make it more speculative and equivocal.

As regards to the background section, is there an example of cultural environment in all social evolution in which individual learning does not require the capacity of identifying role models and relying on their signal/example? It could appear a purely provocative/rhetorical question. However, how could we even imagine a competitive institutional system based on continuous learning and complex incremental paths (scientists compete for extending the frontier of knowledge via priority rewards, to cite Merton who is central in this paper) without cumulative advantages? The authors make a good point in discussing learning and feedback. However, it seems to me that cumulative advantages due to bias and cumulative advantages due to learning and specialisation provide two very much alternative explanations, whereas it seems to me that authors implicitly but straightforwardly take the first line when discussing their findings.

** This particular feedback isn’t entirely clear to us, but it appears to deal with abstract issues that are beyond the scope of this paper.

Figures 1 and 2: the meaning of the distribution in the axis of the publication order is unclear. Please, add a legend.

Figure S1: the scale of axes must be kept similar or the comparison between the two plots is non-intuitive

Tables S1 and S3. I would add an explanation of the range of journals in the legend.

** We redid all of the figures; hopefully it will be clearer. However, we do not think there’s a better suggestion than using “publication order” for the repeat publications in a given journal. We think most readers will understand the figures.

Reviewer #2: This paper provides some very interesting new descriptive statistics on the prevalence and correlates of repeat publishing in economic and general-science journals. My only concern with the paper as currently written is related to the discussion and conclusion sections of the paper, which often make suppositions that read slightly more dramatic than I believe is warranted given the authors results.

I think the authors put it best themselves when they say on pg. 18: "Whether this is due to skill, luck and/or social connections of those repeat authors is an open question."

However, the discussion and, in particular, the conclusion section is written with a rather negative tone highlighting many potential negative consequences of the patterns in the data. As the authors note, while there is evidence of decreasing citations for repeat publishers, those same papers often receive more citations than the other papers in the same journal (by non-repeat publishers). So if editors are maximizing citations, this strategy seems rational. And of course, as the authors note as well, citations may not fully capture important dimensions of paper quality (e.g., introduction of new, diverse viewpoints). But this also implies that the decline in citations amongst repeat-publishers could reflect the repeat-publishers branching out into new domains or research questions, in which case this pattern could still be a sign of "good" strategies by authors and editors. I don't mean to say that the authors' conclusions are wrong, they are absolutely reasonable hypotheses that could rationalize the observed data. My point is to say that there are many other hypotheses (that don't require negligence or biases) that can also rationalize these patterns. Thus, I would appreciate a more even handed approach to the writing in these final sections.

** Agreed. We tempered some of the language in the conclusion. We also made the main finding clearer that repeat authors do tend to perform better than debut authors. While there may be costs of winnowing the pool of contributors to a journal, the incentives presented to gatekeepers as suggested by our research is pretty clear.

These remarks are why I have listed this paper as "partly" supporting the authors conclusions. I think a more balanced discussion of what could be generating these patterns in the data would be worthwhile and convince me that the data more fully supports the hypotheses put forward in the discussion and conclusion sections.

---

## [Decision Letter · Decision Letter 1]

13 Dec 2021

PONE-D-21-19824R1Cumulative Advantage and Citation Performance of Repeat Authors in Academic JournalsPLOS ONE

Dear Dr. Siler,

Thank you for submitting your manuscript to PLOS ONE. After careful consideration, we feel that it has merit but does not fully meet PLOS ONE’s publication criteria as it currently stands. Therefore, we invite you to submit a revised version of the manuscript that addresses the points raised during the review process.

One of the two reviewers continues to point out some issues that I agree you should consider before the article is accepted for publication.

In particular,

1) you should justify more convincingly the choice of juxtaposing three multidisciplinary journals with the whole set of economics journals. In my opinion, it should be more ‘natural’ to present a comparison of two research fields rather that a comparison among a field and three multidisciplinary journals. Please note that in economics it is usual to consider some journals, such as the top5, as ‘generalist’, and it is therefore a bit puzzling to have this kind of general comparison.  I think that readers would benefit of a better discussion of your choice.

2) you should better justify the choice of considering US only authors. I think that you should discuss this as a limitation of your work.

I think that Figure 1-4 should be modified by adopting inside each Figure identical y-scales for the two panels. The use of different scales may confound readers.

As for data availability, I think that the statement you trasmissed on 19 October 2021 (""The data are proprietary and are property of Clarivate Analytics and Leiden University. Data are available for researchers who meet the criteria for access to this dataset. Aggregated data will be available on Figshare upon acceptance of the manuscript. To obtain the bibliometric data in the same manner as authors (i.e. by purchasing them), readers can contact Clarivate Analytics at the following URL: https://clarivate.com/products/web-of-science/contact-us.")  is fully compliant with journal policy. I wonder whether you could consider making the micro data available in some form, for example by replacing the name of authors by a conventional ID. (Please note that this last issue is not an impediment to the publication of the article.)

We look forward to receiving your revised manuscript.

Kind regards,

Alberto Baccini, Ph.D.

Academic Editor

PLOS ONE

Reviewers' comments:

Reviewer's Responses to Questions

**Comments to the Author**

1. If the authors have adequately addressed your comments raised in a previous round of review and you feel that this manuscript is now acceptable for publication, you may indicate that here to bypass the “Comments to the Author” section, enter your conflict of interest statement in the “Confidential to Editor” section, and submit your "Accept" recommendation.

Reviewer #1: (No Response)

Reviewer #2: All comments have been addressed

2. Is the manuscript technically sound, and do the data support the conclusions?

Reviewer #1: Partly

Reviewer #2: Yes

3. Has the statistical analysis been performed appropriately and rigorously? 

Reviewer #1: Yes

Reviewer #2: Yes

4. Have the authors made all data underlying the findings in their manuscript fully available?

Reviewer #1: No

Reviewer #2: Yes

5. Is the manuscript presented in an intelligible fashion and written in standard English?

Reviewer #1: Yes

Reviewer #2: Yes

6. Review Comments to the Author

Reviewer #1: I must confess I am a bit puzzled by the authors’ response. It is evident that they have economized on their revision. I focus here on the most important points on which I would recommend that authors elaborate in the paper.

In my report, I recommended authors to justify their choice of the field selection in their research design, i.e., their choice to compare top economics journals with some top generalists. They simply responded in the correspondence but did not elaborate in the paper. This is a problem as any reader unfamiliar with the hierarchy structure of economics journal could not really get their point and so under-evaluate their research. Please, specify since the intro your selection. I am a social scientist, so I know that economics is an incremental discipline with a strong theoretical mainstream that is particularly defended by those top journals. But, research on cumulative advantages in science is also read by non-social scientists. And, in any case, it is good practice to explain the authors’ research design choices to the reader.

Secondly, the fact that they limited the analysis to US authors in order to control for country as publication trends are country-specific is not specified in the manuscript. They also responded that “US authors account for the majority of econ papers as well as of papers in Science / Nature and PNAS throughout the period”. Well, good to add these points in the paper. I don’t want to question this point so hardly asking the authors if there anywhere in which they were required to use country as a control variable in their estimates, which would be the real point, but again, the reader has the right to know more about their choices.

When requested to tone-down their guess on cognitive bias in editorial decisions, for which they did not have any concrete measurement, they responded that they only speculated about the potential mechanisms to then pick up a favorable citation, e.g., Szenberg and Ramrattan, 2014, where again no concrete measurement to support such claims was provided. This is a quantitative study and so authors know very well how much it is important to support claims with evidence. I would recommend authors to discuss more the limitations of their study by adding a couple of paragraphs on the study limitations. This would add a lot to the paper.

Reviewer #2: (No Response)

7. PLOS authors have the option to publish the peer review history of their article (what does this mean?). If published, this will include your full peer review and any attached files.

Reviewer #1: No

Reviewer #2: No

---

## [Author Response · Author response to Decision Letter 1]

15 Feb 2022

As per point #1 in Dr. Baccini's last decision letter, we added explanations of the contrast between 'generalist' and 'specialist' journals in the manuscript. We believe the breadth of our journals is a unique feature and strength of the manuscript, as it shows how repeat author dynamics occur in very different scholarly contexts. 

As per point #2, we also explained why analyses were limited to US authors. Given linguistic, cultural and institutional differences in universities, disciplines and academic journals, we felt that restricting the analysis to the USA (which was by far the most prolfiic publishing country) was a 'safe' research decision.

Additionally, we changed the figures to ensure the axes are on identical scales.

---

## [Editor Report · Decision Letter 2]

9 Mar 2022

Cumulative Advantage and Citation Performance of Repeat Authors in Academic Journals

PONE-D-21-19824R2

Dear Dr. Siler,

We’re pleased to inform you that your manuscript has been judged scientifically suitable for publication and will be formally accepted for publication once it meets all outstanding technical requirements.

Kind regards,

Alberto Baccini, Ph.D.

Academic Editor

PLOS ONE
---

## [Editor Report · Acceptance letter]

22 Mar 2022

PONE-D-21-19824R2 

Cumulative Advantage and Citation Performance of Repeat Authors in Scholarly Journals 

Dear Dr. Siler:

I'm pleased to inform you that your manuscript has been deemed suitable for publication in PLOS ONE. Congratulations! Your manuscript is now with our production department. 

Kind regards, 

on behalf of

Prof. Alberto Baccini 

Academic Editor

PLOS ONE